# Root Resorptions on Adjacent Teeth Associated with Impacted Maxillary Canines

**DOI:** 10.3390/diagnostics12020380

**Published:** 2022-02-01

**Authors:** Sanja Simić, Predrag Nikolić, Jelena Stanišić Zindović, Radovan Jovanović, Ivana Stošović Kalezić, Aleksandar Djordjević, Vesna Popov

**Affiliations:** 1Department of Orthodontics, Faculty of Medicine, University of Priština in Kosovska Mitrovica, 38220 Kosovska Mitrovica, Serbia; jelena.stanisic@med.pr.ac.rs (J.S.Z.); radovan.jovanovic@med.pr.ac.rs (R.J.); ivana.stosovic@med.pr.ac.rs (I.S.K.); aleksandar.djordjevic@med.pr.ac.rs (A.D.); 2Department of Orthodontics, Faculty of Dentistry, University of Belgrade, 11000 Belgrade, Serbia; predrag.nikolic@stomf.bg.ac.rs; 3Orthodontic Associates Baltimore, Catonsville, MD 21228, USA; vesnapopov77@gmail.com

**Keywords:** impaction, maxillary canine, root resorption, CBCT

## Abstract

Aim: Through the use of CBCT images, many unidentified features of impacted canines can be easily resolved. The potential collision of impacted maxillary canines and adjacent teeth could lead to root resorption (RR). The aim of this study is to examine the prevalence, location and severity of RR on adjacent teeth caused by impacted maxillary canines and the association between the adjacent teeth and the features of maxillary impacted canines on CBCT. Methods: This study examined 89 subjects with 108 maxillary impacted canines, having had no previous orthodontic treatment (mean age: 18.3 ± 4.1 years). The following impacted-canine-related parameters were analyzed on the CBCT images: location; RR levels on adjacent teeth; occlusal line and midline distances of impacted canines; and the angulations of impacted canines to the midline, lateral incisor and occlusal line. Logistic regression was used to evaluate the association between RR and the measured parameters on CBCT. Results: In this study, we found that the majority of our patients with impacted maxillary canines were female (62.5%). Of the total 108 maxillary impacted canines, 60.2% resorbed the adjacent teeth of the affected quadrants. Lateral incisors were the most affected (34.3%). The mean age of subjects with RR was 16.7 ± 3.5 years. The frequency of RR was statistically significantly higher in female subjects (40.4%). Slight RR was the most frequent (30.5%) and the highest incidence noted at the apical third of the root (29.6%). Regarding the impacted maxillary canine angulation to the midline and adjacent tooth, higher values of angulation caused severe forms of RR (*p* < 0.05). Conclusion: The sensitivity of CBCT allows for the accurate diagnosis of the location and the degree of RR, alongside the angulation and distance of impacted canines to adjacent teeth. The association between the linear and angular features of the impacted maxillary canines and RR was confirmed.

## 1. Introduction

The maxillary canines are the second most frequently impacted teeth after the third molars, with general prevalence rates from 1 to 3% [1,2]. However, in comparison with the third molar, the maxillary canines are located in a highly demanding area, both in function and aesthetics [3].

The potential collision of impacted maxillary canines and adjacent teeth could lead to root resorption (RR). RR is a relatively common phenomenon, defined as the progressive loss of cementum and dentine of the affected teeth, resulting in permanent tooth root. Due to a general lack of symptoms, the RR of permanent teeth caused by impacted canine collision has a tendency to be misdiagnosed [4]. RR is typically diagnosed by radiography. In recent years, the techniques used to detect maxillary impacted canines have improved with new advances in medical imaging technologies. Through the use of CBCT images, many unidentified features of impacted canines can be easily resolved [5,6,7].

Furthermore, relative to 2D radiography, CBCT enhances the detection of RR on adjacent teeth, contributes to modifications in treatment plan, and improves confidence in diagnosis and treatment plans [8]. Furthermore, whereas 2D radiographs reveal RR in 30–50% of lateral incisors in individuals with impacted canines, the detection of lateral incisor RR increases by at least another 65% when using CBCT [9].

Today, it is clear that CBCT imaging is an important stage in making a diagnosis and planning treatment procedures for impacted canines. It is important to define the exact position relative to neighboring structures and the inclination of the longitudinal axis of the impacted tooth [10]. The presence or absence of RR determines the optimal treatment strategy. When RR occurs, surgical exposure and orthodontic traction of the canine are carried out [11].

According to recent studies, using CBCT images as a diagnostic method has shown that 48% of ectopic canines caused RR of varying severity [6,12]; up to 70% of impacted maxillary canines cause RR of at least one adjacent tooth [13,14].

RR associated with impacted maxillary canines seems to be a rapid, progressive process that almost always ceases once the impacted canine has been removed from the affected root area [15]. The complexity of the orthodontic traction of impacted maxillary canines is not a risk factor for greater RR of maxillary incisors close to the impaction area [16].

The aim of this study is to examine the prevalence, location and severity of RR on adjacent teeth caused by impacted maxillary canines among untreated subjects. The secondary aim is to corroborate the association between the adjacent teeth and the features of maxillary impacted canines (i.e., location, distance, angulation) on CBCT.

## 2. Materials and Methods

This study included 89 subjects, ages 12.5 to 34.1, with 108 impacted maxillary canines, having had no previous orthodontic treatment. In all subjects, a standard specialist examination was performed to determine the absence of one or both maxillary permanent canines or the persistence of deciduous canines. After clinical and radiographic examinations, those canines were defined as impacted canines in this study. Patients presenting cysts related to the studied impacted canines, as well as patients with supernumerary teeth, or missing lateral incisors or premolars, were excluded from further analysis. The study was carried out by analyzing the cone-beam computed tomography (CBCT) of the maxilla, which the subjects underwent according to our instructions, only for diagnostic purposes and in order to plan upcoming orthodontic treatment.

For every impacted canine, the following parameters were measured on CBCT:(1)Type of impaction (unilateral, bilateral);(2)Sagittal location (labial, palatal or median) using sagittal and coronal CBCT scans (Figure 1a);(3)Vertical location of the cusp tip in relation to the long axis of the adjacent tooth (on sagittal and axial CBCT scans), which was assigned to one of five categories: subdivided into coronal, cervical third, middle third, apical third of the root, or suprapical;(4)Horizontal position of the canine cusp tip; the canine was observed to overlap with adjacent teeth using sagittal or coronal CBCT scans. It was assigned according to Ericson and Kurol [17] (sector 1: canine overlapping by up to half the width of the lateral incisor; sector 2: canine overlapping by over half the width of the lateral incisor; sector 3: canine completely overlapping with the lateral incisor; sector 4: canine overlapping by up to half the width of the central incisor; sector 5: canine overlapping over the midline of the maxilla) (Figure 1b);(5)Distance of the impacted maxillary canine cusp to the midline (measured on axial CBCT scans) (Figure 2a);(6)Distance of the impacted maxillary canine cusp to the occlusal line (measured on CBCT images in the sagittal plane);Figure 1(**a**) Occlusal reference arch–location of impacted maxillary canines in axial plane; (**b**) horizontal position of canine cusp tip: the canine overlap with adjacent teeth in coronal plane (right maxillary impacted canine in sector 2 ad left maxillary impacted canine in sector 4).
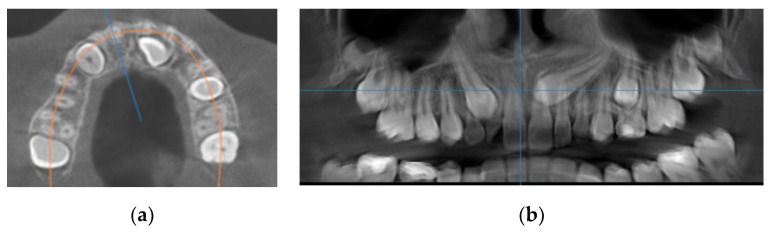
Figure 2(**a**) The distance of maxillary canine cusp to midline (measured on CBCT images in axial plane); (**b**) Angle between long axis of impacted maxillary canine and long axis of adjacent lateral incisor measured on CBCT images in sagittal plane.
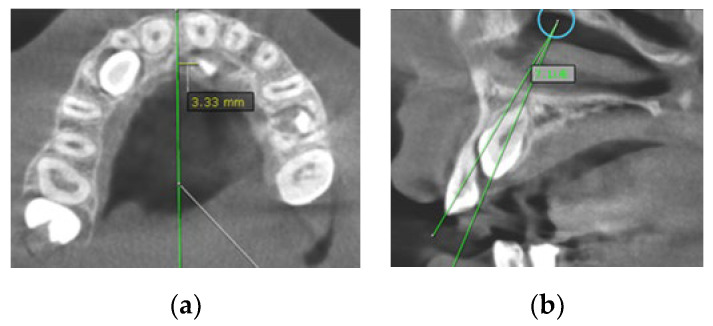
(7)Angle between the longitudinal axis of the impacted maxillary canine and the long axis of the adjacent central/lateral incisor (measured on CBCT images in the sagittal plan) (Figure 2b);(8)Angle between the longitudinal axis of the impacted maxillary canine and the maxillary arch midline (measured on CBCT images in the coronal plan) (Figure 3a);Figure 3(**a**) Maxillary impacted canine angulation to the midline (measured on CBCT images in the coronal plan); (**b**) maxillary impacted canine angulation to the occlusal line and the distance canine cusp to occlusal line (measured on CBCT images in the sagittal plane).
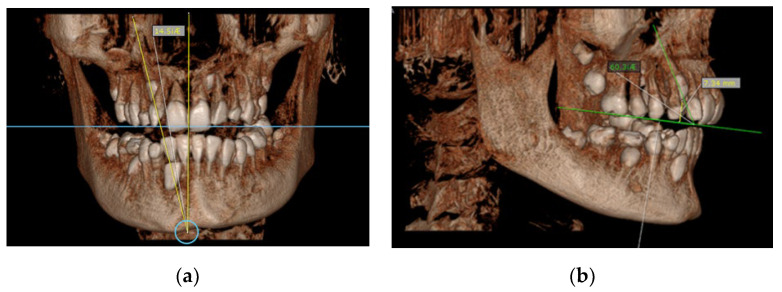
Angle between the longitudinal axis of the impacted maxillary canine and the occlusal line (measured on CBCT images in the sagittal plane) (Figure 3b);(9)RR of the adjacent tooth assessed in the axial plane, using a previously established classification. If RR was suspected, resorption was graded based on the system suggested by Ericson [18] for each tooth into 4 categories: no resorption (intact root surface, the cementum layer may have been lost), slight resorption (resorption up to half of the dentine thickness), moderate resorption (resorption of the dentine midway to the pulp or more, the pulp lining being unbroken), and severe resorption (resorption reaches the pulp). The presence or absence of RR was assessed on 3D MPR views along the long axis of every adjacent root;(10)Localization of RR (cervical, middle or apical third of root).

Logistic regression was used to evaluate the association between resorption and measured parameters by CBCT (Kappa’s coef). Spearman’s correlation test was used to examine the correlation between the parameters measured on CBCT scans.

Mann–Whitney test was used to examine differences between parameters. Statistical significance was set at the level of 0.05.

## 3. Results

In this retrospective study, a total of 89 subjects were included and 108 impacted maxillary canines were analyzed on CBCT scans. The mean age of subjects was 18.3 ± 4.1 years. Of the 89 included patients, 31 were male (34.8%) and 58 (65.2%) female, so there was a statistically significant difference for female patients (*p* < 0.001).

Unilateral impaction was presented in 70 (64.8%) patients and 19 (17.6%) patients presented with bilateral impaction. The analyses of the three-dimensional location revealed that most of the impacted canines were located in the palatal position (80 (74.1%)), 25 (23.1%) were located buccally and only 3 canines (2.8%), in the middle of the alveolar process. Regarding vertical and horizontal localization, impacted maxillary canines were most frequently located in the middle third of the adjacent root (34 (31.5%)) and in sector 3 (39 (36.1%)).

In Table 1, descriptive data regarding the sample are presented. Of the 89 patients in total, RR was found in 47 (52.8%) patients. The mean age of subjects with RR was 16.7 ± 3.5 years. The frequency of RR was statistically significantly higher in female subjects (40.4%). Regarding the sagittal position of the impacted maxillary canines, 13.5% RR was found with labially impacted canines and 39.3% RR with palatally impacted canines. There were no resorptions with mid-alveolar canine impaction. There was a significant difference for RR with palatally impacted canines (*p* < 0.05).

We detected 65 cases of RR (60.2% of the affected quadrants): 19 (17.6%) on the central incisors, 37 (34.3%) on the lateral incisors and 9 (8.3%) on the first premolars. The highest incidence was noted at the apical third of root (29.6% RR): on the central incisors—8.3%, on lateral incisors—15.7%, and on first premolars—5.5%. It was found that 30.5% of identified cases of RR were slight, 18.5% moderate and 11.1% severe. Slight RR was found in 30.5% of incisors and first premolars, with high incidence in the apical third of the root. A high proportion of the involvement of moderate RR (18.5%) was detected in the middle third (10.2%). The cervical third was the least affected (11.1%) (Table 2).

The relationship between RR severity and vertical location is presented in Figure 4.

In the group with RR, the average angle between the impacted canines and midline was 46.7°, and the average angle between the impacted canines and the adjacent lateral/central incisors was 43.1°/40.3°. The average angle between the canine and occlusal line was 43.6°, whereas the average angle in the group with no resorptions was 55.8° (Table 3). Based on the clinical findings in the group of subjects with RR on the incisors, the angulation of the impacted canine to the midline and angulation to the incisor were found to have significantly higher values than in the group of subjects with no resorption. On the other hand, the angulation of the impacted canine to the occlusal line was found to be lower than in subjects with no resorption, and it was statistically confirmed (Figure 2).

There is a significant statistical correlation between the severity of RR on incisors and the angulation of the impacted canine to the incisor, affected by the resorption and angulation of the canine to the midline. More precisely, higher values of these angulations cause severe forms of RR. Regarding the impacted maxillary canine angulation to the occlusal line, no statistical significance with incisor RR was shown.

Therefore, the measured distance from the canine cusp tip to the occlusal line (11.7 ± 3.6 mm) is considered to be a significant parameter, since it shows statistical significance with respect to RR in adjacent teeth; namely, the higher position of the impacted canine causes a more severe degree of RR (Figure 5). The mean values for the distance from canine cusp tip to midline (9.25 ± 4.4 mm) are also a statistically significant parameter concerning the severity of RR.

There was no statistical significance between the vertical position of the crown and the severity of RR, but there was a statistically significant correlation between the severity of RR and the degree of the horizontal overlap of the canine cusp and adjacent teeth, the greater degree of overlap leading to advanced severe resorption. The results for RR in the first premolars are also different. There was statistical significance for the horizontal position of the impacted canine, angulation to midline and distance to occlusal line (Table 4).

## 4. Discussion

Over the years, clinicians have searched for clues that may indicate a high risk for incisor RR associated with impacted maxillary canines. CBCT enables the determination of the exact distance of adjacent teeth; such a relationship is almost impossible to accurately assess on OPT.

CBCT provides more precise information in diagnostic analysis, especially for planning orthodontic and surgical procedures where complications can be expected due to the close relationship between maxillary impacted canine and adjacent teeth.

In our study, as well as in all other studies, maxillary lateral incisors were found to be the most affected teeth, followed by maxillary central incisors. We found similar results in other publications [3,5,19]. Other studies show different results, suggesting that the first premolars are more often resorbed than the central incisors [20]. Walker [3] found 66.7% and Olsen [21] 67.6% resorption of lateral incisors. The study by Rafflenbeul et al. [22] detected more two thirds of RR, and Castro [23] 46% RR in untreated patients, while our results show 60% RR.

All root levels and surfaces of teeth associated with impacted maxillary canine teeth can be resorbed, with different levels of severity [1,2,19,21,22,23,24,25]. Our study supports previous findings, suggesting that slight RR is predominantly located in the apical third of the incisors and premolars. In all adjacent teeth examined, RR presented most often in the apical and middle thirds of the root. Severe RR was the least presented.

Diagnosed RR usually does not change prior to orthodontic treatment, but significantly affects the treatment plan in terms of determining the direction of orthodontic traction. Otherwise, RR existing on the adjacent teeth may become worse when displacing the impacted canine. This predominance is confirmed by all studies excluding patients with past or ongoing orthodontic treatment [15,25,26,27]. The prevalence of moderate and severe RR tends to be higher in most other studies, perhaps because, in cases of past or ongoing orthodontic treatment [22], poor control of the relationship between the canine and the adjacent roots could have worsened the RR that was already present to a lesser extent.

We decided to compare the angles and distances between the impacted maxillary canines causing RR and those that do not cause RR on adjacent teeth. Dagsuyu et al. [13] compared the parameters of the impacted canines on the left and right side. Algerban et al. [28] suggested that impacted canine angulation of >31° to the lateral incisors is a relevant predictor of canine impaction. Meanwhile, Ericson et al. [29] reported a relationship between impacted canine angulation and RR, finding that RR increased by 50% when the impacted canine angulation to the midline exceeded 25° and also when the impacted canine inclination to the long axis of the lateral incisor exceeded 28°. This is confirmed by our research. These differences may simply arise from the different study groups used.

No relationship was found regarding RR and the position of the canines in relation to the cervical, middle or apical part of the lateral and central incisors. The same results are shown in a study by Kalavritinos et al. [14]. Meanwhile, Ericson et al. [5] stated that the mesial position of an impacted canine crown was a relevant predictor of RR, in agreement with our results. No relationship was found regarding RR and the inclination of the canine to the occlusal plane. On the contrary, there is a positive relationship between the appearance of RR and the canine’s distance to the occlusal line. Yu [30] and Sajnani [31] found that the distance of the canine cusp tip to the occlusal line was the most important predictor of maxillary canine impaction and resorption. A more severe RR was apparent when the impacted canine was positioned higher and when the canine crown overlapped the lateral incisor root by a greater area. The risk is greater when the canine is vertically above the lateral incisor root and close to the median palatine suture, suggesting a mechanical blockage by the apex of the lateral incisor [32].

The severity of displacement, as measured by the position of the canine cusp tip in sectors 1–5, was significantly associated with the RR of incisors and first premolars only. The central incisors presented a higher risk of developing RR if the canine was in their immediate vicinity (sectors 4 and 5), whereas the first premolars presented a lower risk of RR if the canine was further away (sector 3,4,5) [22].

Further research will focus on monitoring diagnosed RR during orthodontic treatment, the expansion of impacted maxillary canines, and whether and how they will change. Since there was not a significant number of subjects with RR on the first premolars, we did not compare them with the mentioned parameters. Thus, RR on the premolars will be the matter of some extensive research.

The sensitivity of CBCT compared with panoramic imaging is much higher, allowing accurate diagnosis of the location and the degree of resorptive cavities, which may be critical in changing treatment plans. Although such treatment decisions appear to be a logical clinical outcome with the use of CBCT, the effects of the superior information derived from CBCT images will be of benefit in modifying treatment decisions and determining the threshold of RR. The use of CBCT increases the detection of RR significantly, by eliminating blurring and the overlapping of other teeth.

Delayed treatment of impacted canines and inappropriate force vectors during active orthodontic treatment may lead to the RR of the adjacent teeth, specifically lateral and central incisors. All root levels and surfaces of teeth can be resorbed to different severity. It is clear that CBCT imaging is essential in the diagnosis and treatment planning of orthodontic patients with impacted canines [33,34].

For proper orthodontic mechanics, it is essential to determine the exact position of the impacted tooth relative to the adjacent teeth and surrounding structures. Knowing the inclination of the longitudinal axis of the impacted tooth will allow for improved force vectors. The early diagnosis of “possible” tooth collision is paramount in avoiding RR and can dictate an optimal treatment strategy. When RR occurs, the surgical exposure and orthodontic traction of the canine may be necessary to avoid further RR and aid in orthodontically erupting the tooth. Monitoring during treatment with progressive (additional) CBCT is recommended in some cases to confirm movements.

Further research will focus on the monitoring of diagnosed RR during orthodontic treatment, the expansion of impacted maxillary canines, and whether and how they will change.

## 5. Conclusions

-The prevalence of impacted maxillary canines and RR on adjacent teeth was higher in female subjects.-Resorption of adjacent teeth was 60.2%, and slight RR was the most frequent.-The maxillary lateral incisors were more frequently affected than central incisors.-The sensitivity of CBCT allows the accurate diagnosis of the location and the degree of RR, alongside the angulation and distance of impacted canines to adjacent teeth.-The measured parameters on the canines improve the indication of RR (the higher position of the impacted canine, the higher values for the angulation of canine to the midline/lateral incisor and the greater degree of horizontal overlap leading to the impacted canine cusp with adjacent teeth).-The association between the linear and angular features of the impacted maxillary canine and RR was confirmed.

## Figures and Tables

**Figure 4 diagnostics-12-00380-f004:**
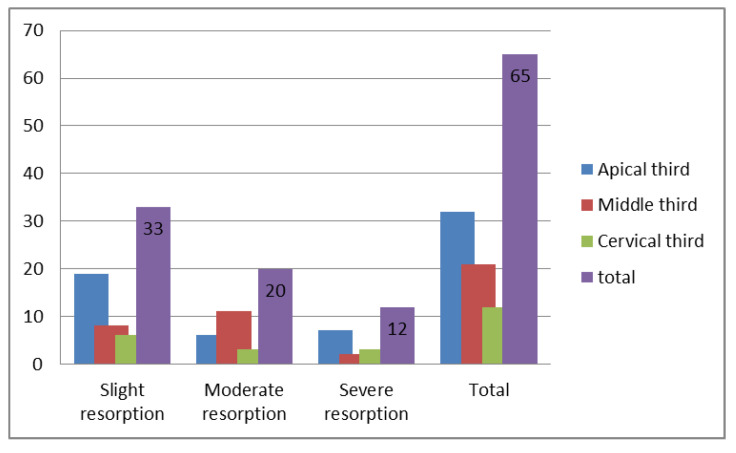
Relationship between severity and vertical location of root resorption.

**Figure 5 diagnostics-12-00380-f005:**
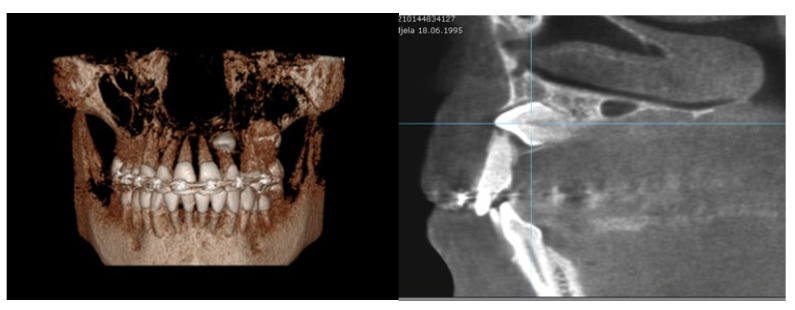
Horizontal position of left permanent maxillary canine associated with severe RR on lateral incisor (3D volumetric scan and sagittal plane).

**Table 1 diagnostics-12-00380-t001:** Distribution of impacted maxillary canine and root resorption.

	Impacted Maxillary Canines		Root Resorptions
**Subjects**	n = 89 (100%)		n = 47 (52.8%)
**Age (mean ± SD)**	18.3 ± 4.1years		16.7 ± 3.5years
**Canines**	n = 108		RR n = 65 (60.2%)
	Male Female	31 (34.8%) 58 (65.2%)	11 (12.4%) 36 (40.4%)
**Canine localization sagittal**	Labial Palatal Median	25 (23.1%) 80 (74.1%) 3 (2.8%)	12 (13.5%) 35 (39.3%) 0
	Unilateral Bilateral	70 (64.8%) 19 (17.6%)	39 (43.8%) 8 (8.9%)
**Canine localization vertical**	Suprapical Apical third Middle third Cervical third Coronal	7 (6.5%) 26 (24.1%) 34 (31.5%) 25 (23.1%) 16 (14.8%)	0 32 (29.6%) 21 (19.4%) 12 (11.1%) 0
**Canine localization horizontal**	Sector 1 Sector 2 Sector 3 Sector 4 Sector 5	27 (25%) 18 (16.7%) 39 (36.1%) 21 (19.4%) 3 (2.8%)	0 (0%) 9 (10.1%) 20 (22.5%) 16 (17.9%) 2 (2.2%)
**Distance to midline (mean)**	9.25 mm ± 4.4 mm		
**Distance to occlusal line (mean)**	11.7 mm, SD ± 3.6 mm		

**Table 2 diagnostics-12-00380-t002:** Prevalence, location and severity of root resorption.

**RR on Adjacent Teeth**	**Central Incisors** **No = 89 (82.4%)** **Yes = 19 (17.6%)**		**Lateral Incisors** **No = 71 (65.7%)** **Yes = 37 (34.3%)**		**First Premolars** **No = 99 (91.7%)** **Yes = 9(8.3%)**		**Total** **No = 43 (39.8%)** **Yes = 65 (60.2%)**
**Location of resorption**							
Apical third		9 (8.3%)		17 (15.7%)		6 (5.5%)	32 (29.6%)
Middle third		7 (6.5%)		11 (10.2%)		3 (2.8%)	21 (19.4%)
Cervical third		3 (2.8%)		9 (8.3%)		0	12 (11.1%)
**Severity of resorption**							
Slight		10 (9.2%)		18 (16.7%)		5 (4.6%)	33 (30.5%)
Moderate		6 (5.5%)		11 (10.2%)		3 (2.8%)	20 (18.5%)
Severe		3 (2.8%)		8 (7.4%)		1 (0.9%)	12 (11.1%)

**Table 3 diagnostics-12-00380-t003:** Difference of impacted maxillary canine angulation in degree between two patient groups.

**Variables**	**Values**	**Resorption on Lateral Incisors**		** *p* ** **Value**	**Resorption on Central Incisors**		** *p* ** **Value**
		**No (n = 71)**	**Yes (n = 37)**		**No (n = 89)**	**Yes (n = 19)**	
**Anglulation canine to midline**	Median (range)	21.4 (12.4−55.6)	46.7 (14.3−86.1)	0.005 *	25.3 (6.1−75.6)	43.5 (17.5−64.6)	<0.001
**Angulation canine to incisor**	Median (range)	23.6 (11.1−48.5)	43.1 (18.1−86.4)	0.004 *	23.1 (8.2−86.2)	40.3 (20.4−61.8)	<0.001
**Anglulation canine to occlusal line**	Median (range)	55.8 (17.4−75.1)	43.6 (22.3−78.8)	0.007 *	59.4 (3.8−78.6)	41.4 (17.4−65.5)	0.004 *

* Mann-Whitney test.

**Table 4 diagnostics-12-00380-t004:** Correlation between angulation, distance and position of impacted canines (with) and severity of root resorptions.

Variables	Values	Severity of Root Resorptions on Lateral Incisors	Severity of Root Resorptions on Central Incisors	Severity of Root Resorptions on First Premolars
**Anglulation canine to midline**	rho	0.293 *	0.426 *	0.235 *
	*P*	0.008	0.000	0.002
**Angulation canine to incisor**	rho	0.300 *	0.411 *	0.165
	*P*	0.007	0.000	0.000
**Angulation canine to occlusal plane**	rho	−0.294	−0.319	−0.185
	*P*	0.008	0.004	0.004
**Horizontal position of canine**	rho	0.291 *	0.580 *	0.320 *
	*P*	0.009	<0.001	<0.001
**Vertical position of canine**	rho	0.197	0.117	0.122
	*P*	0.079	0.301	0.255
**Distance of canine to midline**	rho	0.305 *	0.341 *	−0.174
	*P*	0.008	0.005	0.000
**Distance of canine to occlusal line**	rho	0.307 *	0.320 *	0.287 *
	*P*	0.006	0.004	0.006

* Mann-Whitney test.

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
