# Peer review of "Root Resorptions on Adjacent Teeth Associated with Impacted Maxillary Canines"

_diagnostics, 2022, doi:10.3390/diagnostics12020380_

Round 1

Reviewer 1 Report

This is an interesting study, but I have some major concerns on the design of the study and on data analyses that prevent me to endorse its acceptance at the present stage.

Author Response

Reviewer's comments:

This is an interesting study, but I have some major concerns on the design of the study and on data analyses that prevent me to endorse its acceptance at the present stage.

Author's response to reviewer's comments:

According to your recommendations, we made major corrections in our manuscript.

The introductionis improved to provide sufficient background and include all relevant references.The current version has a different design. Methods are described in details, that you can see in submission. We hope that our work can be published and that our knowledge can be useful to other researchers.

Thank you for recommendations. We hope that the changes are significant and our work can be published and that our knowledge can be useful to other researchers.

Best wishes in New year,

Sanja Simić

Reviewer 2 Report

very interesting inall the discussion and result

Author Response

Reviewer comments:

Very interesting inall the discussion and result.

Authors' response to reviewer comments:

Thanks for the nice comments about our work. We hope that our work can be published and that our knowledge can be useful to other researchers.

Best wishes in New year,

Sanja Simić                                                                                               

Reviewer 3 Report

In this study, the authors investigated the correlation between several parameters of the impacted upper canine and the root resorption of adjacent incisors. This is an interesting and clinically relevant topic. The manuscript is overall well-written, however, the authors are suggested to improve the following aspects:

  1. Abstract: only included the % of RR among the patients, please include the % of the RR among all the canines as well.
  2. Introduction: please add literature on the relationship between the risk of canine impaction with the angulation/position of the canine.
  3. Methods: please add brief explanation of the sector 1-5
  4. Methods: what’s the “occlusal line’ in 6), “occlusal plane”?
  5. Methods: please add a schematic diagram on a CBCT image to show how these parameters are measured
  6. P3 Line 124, how can you run statistics on the percentiles of male and female patients? What method was used?
  7. Table 1: the meaning of n=89 and n=47 wasn’t clear- there should be a top row stating which column is impacted canine and which column is root resorption.

Author Response

Reviewer comments:

In this study, the authors investigated the correlation between several parameters of the impacted upper canine and the root resorption of adjacent incisors. This is an interesting and clinically relevant topic. The manuscript is overall well-written, however, the authors are suggested to improve the following aspects:

  1. Abstract: only included the % of RR among the patients, please include the % of the RR among all the canines as well.
  2. Introduction: please add literature on the relationship between the risk of canine impaction with the angulation/position of the canine.
  3. Methods: please add brief explanation of the sector 1-5
  4. Methods: what’s the “occlusal line’ in 6), “occlusal plane”?
  5. Methods: please add a schematic diagram on a CBCT image to show how these parameters are measured
  6. P3 Line 124, how can you run statistics on the percentiles of male and female patients? What method was used?
  7. Table 1: the meaning of n=89 and n=47 wasn’t clear- there should be a top row stating which column is impacted canine and which column is root resorption.

Authors' response to reviewer comments:

  1. We included the % RR among all canines in the abstract.
  2. In introduction: we add literature on the the relationship between the risk of canine impaction with the angulation of the canine/RR because we have results about these measurements.
  3. Methods: horizontal position of imapacted maxillary impaction is explained in sector 1-5 (Ericson and Kurol);
  1. Methods:“occlusal line” is the better form of the phrase then “occlusal plane”; we change that in 6); The occlusal line touches the incisors and occlusal surfaces of first molar which presented on Figure 3a.
  2. Methods: we add a schematic diagram on a CBCT image to show how these parameters are measured (Figure 1,2,3)
  3. Chi-square test was used to test differences between nominal data (frequencies).
  4. Table 1. is corrected how you suggested, new row is added at the top of the table.

Thanks for the nice comments about our manuscript. We hope that our work can be published and that our knowledge can be useful to other researchers.

Best wishes in New year, 

Sanja Simić                                                           

Round 2

Reviewer 1 Report

I think that in this version your work can be published.

Best regards

Reviewer 3 Report

Thank you for revising the manuscript. The quality of the manuscript has been improved.